# Assessing Fracture Toughness and Impact Strength of PMMA Reinforced with Nano-Particles and Fibre as Advanced Denture Base Materials

**DOI:** 10.3390/ma14154127

**Published:** 2021-07-24

**Authors:** Abdulaziz Alhotan, Julian Yates, Saleh Zidan, Julfikar Haider, Nikolaos Silikas

**Affiliations:** 1Division of Dentistry, School of Medical Sciences, University of Manchester, Manchester M13 9PL, UK; julian.yates@manchester.ac.uk (J.Y.); j.haider@mmu.ac.uk (J.H.); nikolaos.silikas@manchester.ac.uk (N.S.); 2Dental Health Department, College of Applied Medical Sciences, King Saud University, Riyadh 11454, Saudi Arabia; 3Department of Dental Materials, Faculty of Dentistry, Sebha University, Sebha 18758, Libya; saleh_0072002@yahoo.co.uk; 4Department of Engineering, Manchester Metropolitan University, Manchester M1 5GD, UK

**Keywords:** PMMA, ZrO_2_ nanoparticle, TiO_2_ nanoparticle, E-glass fibre, fracture toughness, impact strength

## Abstract

*Statement of Problem:* Polymethyl methacrylate (PMMA) denture resins commonly fracture as a result of the denture being dropped or when in use due to heavy occlusal forces. *Purpose*: To investigate the effects of E-glass fibre, ZrO_2_ and TiO_2_ nanoparticles at different concentrations on the fracture toughness and impact strength of PMMA denture base. *Materials and Methods:* To evaluate fracture toughness (dimensions: 40 × 8 × 4 mm^3^; *n* = 10/group) and impact strength (dimensions: 80 × 10 × 4 mm^3^; *n* = 12/group), 286 rectangular tested specimens were prepared and divided into four groups. Group C consisted of the PMMA specimens without any filler (control group), while the specimens in the remaining three groups varied according to the concentration of three filler materials by weight of PMMA resin: 1.5%, 3%, 5%, and 7%. Three-point bending and Charpy impact tests were conducted to measure the fracture toughness and impact strength respectively. Scanning Electron Microscope (SEM) was utilised to examine the fractured surfaces of the specimens after the fracture toughness test. One-way analysis of variance (ANOVA) followed by Tukey post-hoc tests were employed to analyse the results at a *p* ≤ 0.05 significance level. *Results*: Fracture toughness of groups with 1.5 and 3 wt.% ZrO_2_, 1.5 wt.% TiO_2_, and all E-glass fibre concentrations were significantly higher (*p* < 0.05) than the control group. The samples reinforced with 3 wt.% ZrO_2_ exhibited the highest fracture toughness. Those reinforced with a 3 wt.%, 5 wt.%, and 7 wt.% of E-glass fibres had a significantly (*p* < 0.05) higher impact strength than the specimens in the control group. The heat-cured PMMA modified with either ZrO_2_ or TiO_2_ nanoparticles did not exhibit a statistically significant difference in impact strength (*p* > 0.05) in comparison to the control group. *Conclusions*: 1.5 wt.%, 3 wt.% of ZrO_2_; 1.5 wt.% ratios of TiO_2_; and 1.5 wt.%, 3 wt.%, 5 wt.%, and 7 wt.% of E-glass fibre can effectively enhance the fracture toughness of PMMA. The inclusion of E-glass fibres does significantly improve impact strength, while ZrO_2_ or TiO_2_ nanoparticles did not.

## 1. Introduction

For some people, replacing missing natural teeth using artificial substitutes, e.g., acrylic dentures, is vital for psychological and physical health [1,2]. Polymeric materials are commonly utilised in denture bases as they represent a more practical and affordable alternative to dental implants [3]. Of the available polymers, PMMA denture resin is often selected for fabrication purposes on the basis that it offers enhanced clinical performance in comparison with the alternative options [4,5]. Polymethyl methacrylate (PMMA) resin has many useful properties that make it popular for dentures. However, it is associated with a number of drawbacks [1,4,6,7]. The primary challenge associated with most acrylic denture bases is that fractures gradually develop during the first few years of use [8]. These fractures typically result from exposure to high mastication forces. However, a fracture may develop after a denture has been dropped on a hard surface [6,9,10,11]. The majority of fractures appear in the midline of the denture, a phenomenon that is particularly common in maxillary dentures [11,12]. The factors that can contribute to the development of a fracture include functional stresses, the habits of the user, the way the denture is handled, the porosity of the resin, the presence of thin areas within the denture bases, and the formation of cracks [2,9,13]. This difficulty means that resins used to build dentures must have good impact resistance and be able to sustain high-impact forces without cracking [2,9,13].

The failure of acrylic dentures when in use is a serious issue [6,11,14]. Therefore, significant effort has been invested in developing new materials that can resist fractures and, thereby, strengthen PMMA dentures [4,11,14,15]. One means by which it is possible to strengthen PMMA dentures involves modifying its chemical structure through incorporating crosslinking agents or copolymerizing the material with rubber to enhance impact strength [16]. However, this approach reduces flexural strength and fatigue resistance [11]. An alternative method that is frequently employed to strengthen the PMMA denture bases is to include inorganic fillers, such as nanoparticles and fibres, and existing research suggests that they can represent an effective means of enhancing the mechanical properties of PMMA dentures [4,7,14,17]. To date, a range of nanoparticles has been introduced to the polymer matrix, including oxides of aluminium (Al_2_O_3_) [18], silicon (SiO_2_) [19], titanium (TiO_2_) [20,21] and zirconium (ZrO_2_) [14,22]. Carbon nanotubes (CNTs) and graphene-silver nanoparticles have also been found to successfully reinforce PMMA properties due to their exceptional mechanical properties [23,24]. However, the colour of PMMA specimens reinforced with either or both materials (CNTs/Gr-Ag) is not aesthetically pleasing compared to pure PMMA, even at low ratios [24,25]. Nanoparticles have numerous well-known inherent characteristics in terms of shape, size, and composition, and the extra strength they can offer when employed to reinforce polymers [19,26]. Using titanium and zirconium allows for biomechanical and safety standards to be met and research has shown that they are highly suitable materials for indirect dental restoration [22,27,28]. Either of these materials displays a high level of strength, allowing for the endurance of in-use occlusal force; have a high level of biocompatibility, and offer no side effects [28,29,30]. The findings of previous research revealed that the ratio of filler used needs to be low enough to ensure it is embedded in the resin [31]. Furthermore, the nanoparticles should be suitably small to form a homogenous mixture with the resin matrix that facilitates polymer chain movement by enabling nanoparticles to fill any pores that have developed between polymer particles [28]. A study by Saleh et al. [32] observed that an increase in filler content, reduced the fracture toughness and impact strength of PMMA/ZrO_2,_ especially at the 10 wt.% ZrO_2_ concentration. Zhang et al. [22] also concluded that the inclusion of 1 wt.% and 2 wt.% silanised ZrO_2_-ABW/PMMA nanocomposites improved the flexural strength of the material. However, when the nano-ZrO_2_ ratio was further increased, from 2 wt.% to 4 wt.%, a reduction in the flexural strength of the material was observed [22]. Increasing the nanoparticles concentration beyond a certain level fosters more filler-to-filler than filler-to-matrix interactions; thus, agglomeration may represent a point of stress concentration that may generate non-uniform stress distribution [31].

Aramid and carbon/graphite fibres have been used to reinforce the mechanical properties of PMMA denture base [33,34]; however, they have been found to be difficult to polish and are not aesthetically pleasing [16]. Thus, their application has been limited to locations where aesthetics do not represent a significant consideration [17]. Glass fibre is the most popular among the available fibres as it is easy to manipulate, bonds well with resin matrices, has an acceptable aesthetic appearance, and has excellent mechanical properties [4,7,14]. Fibre can clump together when combined with polymer where the glass fibre content is higher [16]. In addition, the porosity of the material rises as a result of the formation of a void within the composite [16]. Thus, researchers typically recommend that low fibre content is used [17,35].

Fracture toughness is used to determine the extent to which materials are resistant to the propagation of flaws under an applied load [13,36]. It is an indication of how much stress a material can sustain before a crack is initiated and propagated through the material [9,37,38]. One approach that is commonly used to determine the fracture toughness of denture bases is to load a sharply notched beam in three-point bending [9,39]. If a denture base material has a high fracture toughness, it has the capacity to resist cracking [13,40]. On the contrary, if the fracture toughness is low, the material is brittle and may crack easily [7]. Therefore, fracture toughness is an important consideration when aiming to fabricate dental materials that offer longevity [7,37], and many studies have focused on methods of increasing the fracture toughness of the PMMA resin dentures [4,6,32,36,41]. Impact tests are frequently performed to simulate experimental fractures and identify the amount of energy that acrylic denture resin can absorb before fracturing [2,13,42]. Two standard methods are used to assess impact strength: the Charpy test and the Izod test. Either of these methods can be employed on a specimen that has a precisely cut notch ensuring that it will fail at a specific point, which allows for notch sensitivity or material brittleness assessment [2,9].

Filler-reinforced composites are typically produced from filler-reinforced polymer matrices [4]. The way in which the reinforced filler behaves will vary according to a range of factors. The strength of the chemical bond between the polymer matrix and the reinforcing filler is of utmost importance [22,32]. One way by which this can be enhanced is through the use of a silane coupling agent that increases the bond [32]. It is also important that the reinforcing filler is uniformly dispersed and impregnated in the matrix to prevent the development of areas of stress concentration that ultimately weaken the resin’s mechanical properties [22]. The findings of various studies have indicated that the use of fibres or nanoparticles to reinforce conventional, heat-cured denture bases enhances mechanical properties such as impact strength, flexural strength, and fracture toughness [10,18,43]. However, there is a lack of literature describing systematic studies with regards to the effect the incorporation of nano-TiO_2_, E-glass fibre, and nano-ZrO_2_ has on the fracture toughness and impact strength of heat-cured PMMA denture bases. Although a variety of materials can potentially be used to reinforce dentures, there is a need to better delineate which materials deliver the optimal mechanical properties. As such, further studies are required to identify which materials, and at what concentrations, best enhance the lifespan and performance of the heat-cured PMMA dentures.

This study assessed the effect of systematically varying the concentration (1.5 wt.%, 3 wt.%, 5 wt.%, and 7 wt.%) of ZrO_2_, TiO_2_ nanoparticles, and E-glass fibres on the fracture toughness and impact strength of the heat-cured PMMA with an underlying intention of identifying the optimum concentration of these reinforcement materials within denture bases. The null hypotheses were that the incorporation of filler would not have a significant influence on (i) the fracture toughness and (ii) the impact strength of PMMA acrylic resin specimens. The hypothesis was tested by comparing two groups: one using reinforced acrylic resins, and the other non-reinforced acrylic resin.

## 2. Material and Methods

### 2.1. Materials

An overview of the materials that were employed in this study is presented in Table 1. Conventional heat-polymerised acrylic resins consist of liquid methyl methacrylate (MMA) monomer and polymethylmethacrylate (PMMA) powder. In addition to using ZrO_2_, TiO_2_ nanoparticles, and silanised E-glass fibre as the filler materials, a silane coupling agent (3-Trimethoxysilyl propyl methacrylate) and ethanol were also utilised.

### 2.2. Specimen Preparation

The control group contained specimens that consisted of pure, heat-cured PMMA. The three experimental groups contained specimens reinforced with ZrO_2_, TiO_2_, and E-glass fibres. Each of the three experimental groups were further divided into four subgroups in accordance with the weight of filler incorporated in the sample: 1.5 wt.%, 3 wt.%, 5 wt.%, and 7 wt.% (Table 2). The specimens were prepared following the manufacturer’s instructions, previous studies [22,32,44,45] and ISO standards [46,47].

#### 2.2.1. Surface Modification of Nanoparticles

To enhance the chemical bond between the acrylic resin matrix and the nanoparticles, the surface of each of the nanoparticles was altered using the silane coupling agent (γ-MPS) before being mixed with PMMA. Firstly, 15 g of each of the nanoparticles was separately mixed with 70 mL of ethanol in a 100 mL plastic container for 20 min using a mixer (DAC 150.1 FVZK, High Wycombe, Buckinghamshire, UK) at a speed of 1500 rpm. This helped to ensure that the particle surfaces had been sufficiently cleaned before being evenly coated with ethanol solution. After that, 0.45 g (3 wt.%) of γ-MPS was incorporated into the resultant suspension of ethanol and nanoparticles, and a uniform mixture was achieved through the use of a magnetic stirrer at 200 rpm over 2 h at room temperature. The suspension was subsequently refluxed over a duration of 4 h at 50 °C. After the completion of the reaction, the mixture was cooled, divided into two equal parts, and poured into 50 mL plastic tubes that were subsequently sealed with a plastic lid. The tubes were then centrifuged (Heraeus, UK) and rotated for 20 min at a speed of 4500 rpm and a temperature of 23 °C. The clear supernatant (which contained the separated ethanol) was decanted, leaving sediment that consisted of the nanoparticle silanised with γ-MPS. Perforated aluminium foil was subsequently used to cover the plastic tubes before they were placed into a Genevac machine (Genevac EZ-2 series, SP Scientific Company, Ipswich, UK) for drying over 3 h at a temperature of 50 °C. This process allowed the solvent to evaporate, leaving the silanised nanoparticles ready for mixing with the matrix.

#### 2.2.2. Dispersion of Filler with the PMMA/MMA

The quantities of the silanised fillers and the heat-polymerised powder were measured using an electronic balance (Ohaus Analytical, Parsippany, NJ, USA), after which the silanised ZrO_2_ nanoparticles were combined with the MMA monomer and combined in a speed mixer for 10 min at a speed of 1500 rpm. This process resulted in the generation of the modified monomer, which was subsequently combined at a ratio of 10 mL modified monomer to 21 g PMMA powder in accordance with the manufacturer’s instructions. The ultimate ratio of monomer to mixing acrylic resin/filler was 5.7 mL monomer to 12 g powder/filler in each mould. The entire process was repeated for the TiO_2_ nanoparticles.

The monomer-powder ratio of 5.7 mL to 12 g was also employed to combine the PMMA/MMA with the silanised E-glass fibres. However, the chopped E-glass fibres were moistened using 4.3 mL of MMA. The fibres were initially closely compacted. As such, they were carefully redistributed by hand to ensure thorough saturation. The PMMA powder weighed to 0.400 g was then mixed with the liquid for a duration of 10 s. This process was repeated six times to ensure that the materials had fully combined. The resulting solution was then stirred for a further 2 min to ensure that the E-glass chopped fibres had adequately embedded in the MMA matrix and were distributed effectively throughout the liquid. In the final step, the remaining resin powder (9.6 g) was added to the E-glass fibre-PMMA mixture, and the remaining monomer (MMA) liquid (1.4 mL) was added.

In both cases, the PMMA/MMA and filler mixtures were stirred with a spatula for approximately one minute to ensure the powder and filler were sufficiently moistened with the monomer. The mixtures were placed in an appropriate container to ensure the monomer could not be evaporated out. After being store at room temperature for 20 min, the mixture had developed a dough-like consistency. It was subsequently shaped by hand into a brass mould that had been pre-treated with a thin layer of a separating medium to ensure that the dough would not affix to the mould during the process of polymerisation. The mould was designed to produce samples for the fracture toughness test and contained a cavity of a dimension of 40 mm (l) × 8 mm (w) × 4 mm (d). Another mould was used to produce samples for the impact strength test; however, the dimensions, in this case, were 80 mm (l) × 10 mm (w) × 4 mm (d). After the dough had been shaped into the mould, it was sealed and placed into a hydraulic press (Sirio Dental, Meldola (FC), Italy). Pressure was applied and gradually increased to reach 1500 Psi. The mould was subsequently clamped to ensure a tight sealing before the mould and clamp were submerged under water in a curing unit (Wassermann Dental-Maschinen GmbH, Hamburg, Germany). Initially, the specimen was cured by gradually raising from room temperature to 74 °C over a period of 90 min, before it was further cured over a duration of 30 min at 95 °C. Once the polymerisation cycle had been completed, the mould was left to cool at room temperature for 30 min to reduce the risk of the specimen becoming warped or stressed upon opening. The specimens were then removed from the mould and finished with 400 and 600 grit silicon carbide emery papers (Buehler Ltd. Esslingen, Germany) before being polished using a lapping machine (MetaServ 250, Buehler Ltd., Esslingen, Germany).

### 2.3. Mechanical Measurements Procedures

#### 2.3.1. Fracture Toughness Measurement

Specimens (*n* = 10) of dimensions 40 (l) × 8 ± 0.2 (h) × 4 ± 0.2 mm^3^ (w) were prepared. After the specimens had cured, a motorised diamond sawing blade (0.5–1.0 ± 0.2 mm-thick) was used to create an initial sharp notch of a depth of 3.0 ± 0.2 mm in the middle of the specimens. This resulted in single-edge notched specimens. To form a pre-crack in the tested specimens, a sharp scalpel was then positioned at the end of the initial notch, and hand pressure was applied to extend the notch to a depth up to 0.1 to 0.2 mm. The length of the notch in each specimen was examined using a stereomicroscope (EMZ-5; Meiji Techno CO., Saitama, Japan). The specimens were then placed in an incubator at 37 ± 1 °C for 7 days. Each single edge-notched sample was tested for fracture toughness through the use of a universal testing machine (Hounsfield H10KS, Tinius Olsen, Horsham, PA, USA) that subjected the samples to 3-point bending configurations with a 500 N load cell, a span length of 32 ± 0.1 mm, and a crosshead speed of 1 mm/min (ISO 20795-1:2008). The apparatus was configured by employing the notch depth of 3 mm and the mean values for thickness and width of the specimens. The specimens were taken out of the water and dried for an hour at room temperature before being tested. They were then placed on edge on the testing rig’s support. The specimen notch was placed diametrically opposite the load-plunger at the centre of the span, as presented in Figure 1. Each beam specimen underwent the application of a central load until the fracture point of the specimen was achieved. To minimise measurement bias, all the specimens in all the experimental groups underwent testing on the same day in the same environmental conditions. The maximum load (*P*) creating a fracture was employed to calculate the fracture toughness (*K_IC_*) in MPa.m^1/2^ as per Equation (1).
(1)KIC=3PL2BW32×Y
where, *P* is the peak load of fracture in Newtons (N), *L* is the span length (mm), B is the specimen width (mm), *W* is the specimen height (mm). The calibration function for the given geometry, *Y* was calculated using Equation (2).
(2)$$Y=[1.93\times {\left(\frac{a}{w}\right)}^{\frac{1}{2}}-3.07\times {\left(\frac{a}{w}\right)}^{\frac{3}{2}}+14.53\times {\left(\frac{a}{w}\right)}^{\frac{5}{2}}-25.11\times {\left(\frac{a}{w}\right)}^{\frac{7}{2}}+25.80\times {\left(\frac{a}{w}\right)}^{\frac{9}{2}}]$$
where *a* represents the notch depth.

#### 2.3.2. Impact Strength Measurement

Twelve specimens with dimensions of 80 mm long, 10 mm wide, and 4 mm thick were fabricated for each tested group. Once the specimens were cured, each specimen was given a Type A notch, a V-shaped notch across the middle, as outlined in ISO 179-1:2000 [47]. This notch was cut by employing manual notch cutting on edge to 2.2 ± 0.1 mm, leaving a residual depth below the notch of 7.8 ± 0.1 mm, with the notch base having a radius at a 45° angle. These specimens were stored in their groups at 37 °C in distilled water for 7 days; they were then placed in a different container full of distilled water at 23 °C for an hour before testing commenced. The impact strength was assessed using a Charpy method with a 0.5 Joule (J) pendulum (Zwick/Roell Z020 Leominster, UK), as specified in ISO 179-1:2000. Each specimen was horizontally placed on the testing instrument supports, which had a span of 40 ± 0.2 mm. The notch was placed facing away from the point at which the pendulum would impact. To fracture the specimen, the 0.5 J pendulum was then released at the mid-span of the specimen at the opposite end to the notch. The results were subsequently digitally recorded as $\;E_{c}$. $a_{c}N$ (Charpy notched impact strength), described in this research as Type A, is the cross-sectional area of the specimens at the notch. Charpy impact strength in kJ/m^2^ for the notched specimens with Type A notches was calculated by employing Equation (3).
(3)$$a_{c}N=\frac{E_{c}}{hb_{A}}\times 10^{3}$$
where, $E_{c}$ is the corrected energy absorbed in the breaking of the test specimen (Joules), *h* is the specimen thickness (mm) and, $b_{A}$ is the width remaining at the *A* notch tip (mm).

### 2.4. Scanning Electron Microscopy (SEM) Analysis

The fractured surface of the specimens from different groups was examined at the point of loading after the fracture toughness test under an SEM (Carl Zeiss Ltd., 40 VP, Smart SEM, Cambridge, UK). To assess the extent to which the fillers had adhered to the resin matrix, the existence of any defects, and the porosity of the samples. After the specimens had been mounted onto aluminium stubs, they were sputter-coated with a thin layer of gold. A secondary electron detector at an acceleration voltage of 2.0 kV was employed to generate SEM visualisations at various magnifications.

### 2.5. Statistical Analysis

The statistical software (SPSS statistics version 25, IBM, New York, NY, USA) was used to statistically assess the fracture toughness and impact stress data for each reinforced group in comparison to the PMMA acrylic resin control group. The outcomes of the Shapiro-Wilk and Levine tests indicated that the *p*-values were not of statistical significance. This indicated that there was a normal distribution of the data and homogeneous variance. To identify whether any subgroup data variations were significant in comparison to the control group, a one-way analysis of variance (ANOVA) was carried out at *p* < 0.05 significance level. A Tukey post-hoc test was then performed as a means of identifying any variations that could be seen across the subgroups in a given group.

## 3. Results

### 3.1. Fracture Toughness

The means and standard deviations of fracture toughness are outlined in Figure 2 and Table 3. The results of ANOVA showed that there was a significant difference (*p* < 0.05) in fracture toughness between the groups that contained filler/fibre and the unfilled group, C (1.42 ± 0.06 MPa.m^1/2^). The fracture toughness of the specimens in the group that was filled with a 3 wt.% ZrO_2_ nanoparticle significantly increased (1.75 ± 0.12 MPa.m^1/2^) by 23.2% in comparison with the fracture toughness of the control group and thus, achieved the highest mean fracture toughness among the reinforced groups. Furthermore, the mean fracture toughness was statistically (*p* < 0.05) higher in Groups Z1.5 and T3, and all E-glass fibre-modified groups in comparison to the control group. The mean fracture toughness in the remaining reinforced groups was increased slightly higher than the control group; however, the differences were not statistically significant (*p* > 0.05).

In terms of the specimens in the ZrO_2_-reinforced nanoparticles, fracture toughness of Groups Z1.5 and Z3 improved significantly compared to the control group but the Groups Z5 and Z7 showed improvement without any statistical significance. No significant differences within the ZrO_2_ subgroups were reported. Overall, the fracture toughness increased significantly at 1.5 wt.% and 3 wt.% ZrO_2_ (1.61 ± 0.15 and 1.75 ± 0.12 MPa.m^1/2^) nanoparticles before dropping gradually when the concentrations of ZrO_2_ were 5 wt.% and 7 wt.% (1.56 ± 0.13 and 1.53 ± 0.12 MPa.m^1/2^).

The group filled with a 3 wt.% of TiO_2_ nanoparticle (1.7 ± 0.16 MPa.m^1/2^) exhibited a significant increase in fracture toughness in comparison to the other TiO_2_ groups and the control group. No significant differences were found between the Groups T1.5/T5, T1.5/T7, or T5/T7.

The findings revealed that increasing the amount of E-glass fibre concentration by 1.5 wt.%, 3 wt.%, 5 wt.%, and 7 wt.% percentage in PMMA denture resin significantly enhanced the mean fracture toughness by 11.97%, 12.7%, 13.4% and 21.13% respectively, in comparison to the control group. However, the increases that were observed in fracture toughness among the E-glass fibre groups were not significantly different, with the exception observed between Groups E7 and E1.5.

### 3.2. Impact Strength

The mean impact strength and standard deviations are presented in Table 3 and Figure 3 for all groups. The ANOVA test results demonstrated that the specimens in the E-glass fibre groups exhibited a significantly (*p* < 0.05) higher mean impact strength than those in the control group. However, the heat cured PMMA that was modified with either ZrO_2_ or TiO_2_ nanoparticles did not exhibit a statistically significant difference (*p* > 0.05) in impact strength compared to the control group.

All ZrO_2_-reinforced groups had an insignificantly higher impact strength than the control group. Generally, the impact strengths fluctuated, increasing in Group Z1.5 and Group Z3 to 3.92 ± 0.37 and 3.98 ± 0.55 kJ/m^2^ respectively, and then decreasing in Groups Z5 and Z7 to 3.81 ± 0.36 and 3.77 ± 0.51 kJ/m^2^, without any significant differences between them.

In terms of the TiO_2_ reinforced groups, the specimens in Groups T1.5 and T3 had insignificantly higher mean impact strength than those in the control group by 1.1% and 5.8%, respectively. However, reduced values were observed in Groups T5 and T7 by 4.42% and 7.73% respectively. No significant differences in the mean impact strength across the specimens in the TiO_2_ groups were observed. Overall, the specimens in the TiO_2_ group follow a similar trend as in the ZrO_2_ but impact strength values were slightly lower.

The groups in which 1.5 wt.%, 3 wt.%, 5 wt.%, and 7 wt.% E-glass fibre were introduced into heat-cured PMMA acyclic exhibited a higher impact strength by 4.7%, 21%, 39.5%, and 51.1%, respectively. These increases were all statistically significant, with the exception of the 1.5 wt.%. A significant difference (*p* > 0.05) was found between Groups E1.5/E5, E1.5/E7, E3/E5, and E3/E7. However, there were no significant differences in impact strength between Groups E1.5/E3 or E5/E7.

### 3.3. Fractured Specimen Analysis

The control specimen exhibited a smooth surface and small pores on the SEM micrograph. This indicated that it had a brittle mode of fracture (Figure 4A). The 7 wt.% ZrO_2_ nanoparticle specimen showed fracture surfaces that were less brittle fracture due it their comparatively irregular rough surface in comparison to the specimens in the control group as seen in (Figure 4B). Small voids and minor agglomerations were observed on the fracture surface of the specimens in the 7% TiO_2_ nanoparticle group (Figure 4C), which is indicative of a poor level of interaction between the TiO_2_ or ZrO_2_ particles and the PMMA matrix. The fracture surface of the E-glass fibres specimen presented a good level of bond between the matrix and the fibres. No gap formation was observed around the fibres (Figure 4D). This indicated that the fibres were more effectively impregnated within the PMMA matrix.

## 4. Discussion

The extent to which denture base material can resist crack propagation and fracture has a direct impact on how it performs over a long-term basis [7,9]. Fracture toughness can be measured by the critical stress intensity factor (*K_IC_*). The *K_IC_* gives valuable insights into the risk of crack propagation, which represents the most common cause of acrylic fracture [32,36]. The single-edged-notched (SEN) specimen test was selected as a means of measuring the fracture toughness in the current study as it consistently yielded reliable and valid results in previous research [36]. However, because of variations in the width and depth of single-edged-notched specimens used in previous studies, it can be difficult to draw comparisons between their respective findings [48].

There is a requirement for denture base materials to endure high masticatory loads or impacts [11]. As such, manufacturers are consistently seeking methods of developing stronger and more durable denture base materials [10]. The impact test represents another approach by which it is possible to predict the fracture resistance of a given material [9]. Impact strength and fracture resistance can be influenced by a variety of factors, including the geometry and position of the specimen, the material used, temperature, stress concentrations, and fabrication variables [11,43]. Hence, there is a need to measure the fracture toughness and impact strength of denture bases as a means of predicting the extent to which they will be able to withstand everyday use and, prove to have longevity [49].

The aim of the current study was to assess a method of enhancing the impact strength and fracture toughness of PMMA heat-cured denture resin through the incorporation of salinised nano-ZrO_2_, nano-TiO_2_, and E-glass fibres at different concentrations. The findings revealed that the hypotheses could be rejected on the basis that incorporating filler materials into the PMMA significantly influenced both the impact strength and the fracture toughness of the PMMA for certain filler concentrations.

The findings of the current study revealed that there was a direct positive correlation with the E-glass fibre content to the impact strength (R^2^ = 0.994) or to the fracture toughness (R^2^ = 0.901) of PMMA denture bases. Furthermore, there was mixed correlation between the nano-ZrO_2_ or nano-TiO_2_ content and impact strength or fracture toughness. In general, an increase in concentration from 1.5 wt.% to 3 wt.% showed increasing trends in the impact strength or fracture toughness while for 5 wt.% to 7 wt.% the trends were opposite. The decreasing trends at higher concentrations could be attributed to many factors including the risk of filler agglomeration, the extent of the additive distribution within the PMMA, and the formation of an interfacial layer between the PMMA matrix and the additives [28]. 3-Trimethoxysilyl propyl methacrylate silane (γ-MPS) can enhance the strength of a bond between a filler material and a given acrylic matrix by nurturing the formation of covalent bonds and a hydrophobic surface [43]. As the nanoparticles offer a large surface area that generates a high level of surface energy, they exhibit a strong propensity to aggregate [3]. If such aggregation occurs, the chemical interaction between the base PMMA and the nanoparticles may reduce [3]. As such, a 3 wt.% silane coupling agent was employed in the current study with the underlying intention of increasing the chemical adhesion between the surface of the filler and the PMMA as a means of producing functional groups [32].

The outputs of this study revealed that the fracture toughness values of the specimens in Groups Z1.5, Z3, and T3 were significantly higher (*p* < 0.05) than those in the control group, C. In addition, all the other remaining ZrO_2_ and TiO_2_ nanoparticle groups showed higher fracture toughness than the control group. This was demonstrated by the brittle fashioned fracture, as observed in the pure PMMA specimen evident on the SEM image (Figure 4A). The highest level of fracture toughness (1.75 ± 0.12 MPa.m^1/2^) was obtained 3 wt.% ZrO_2_. These findings were in agreement with those published by Nejatian et al. [50], who evaluated the correlation between different filler weights—specifically, TiO_2_ of 25 μm, ZrO_2_ of 25 μm, and glass flakes of 350 nm with and the fracture toughness of PMMA. They found that incorporating filler materials at certain concentrations within PMMA did have a positive impact on the fracture toughness of the denture base material. In an alternative study, Asar et al. [43] concluded that modifying heat-cured PMMA with TiO_2_ of 6.9 μm average size, Al_2_O_3_ of 12.4 μm average size, and ZrO_2_ of 8.6μm average size microparticles significantly increased the fracture toughness of PMMA. The rise in fracture toughness that was observed in the current study might be attributable to the size of the nanoparticles involved; i.e., small nanoparticles of around 50 nm ZrO_2_, 25 nm TiO_2_, and 15 μm E-glass fibre were incorporated into 90 μm acrylic powder. In addition, the preparation method involved ensuring that the nanoparticles were fully dispersed and distributed throughout the PMMA through the use of a speed mixer [28,51]. This can serve to reduce the risk of agglomeration within the composites [32]. A further explanation of the increase in fracture toughness could be attributed to the stress transfer between the weak polymer matrix and the hard filler particles [28]. This, the fracture toughness was improved as a result of the uniform distribution of low levels of nanoparticles and stronger bonding between the silane treated nanoparticles and the PMMA matrix. When the concentration of the nanoparticles was increased beyond 3%, a reduction in fracture toughness was typically observed. This reduction could be attributed to the degree of filler agglomeration [32,43] and the fact that void may form points of stress concentration within the polymer matrix that interrupt the uniformity of the stress distribution and, as such, reduce the level of interfacial bonding between the matrix and the nanoparticles [17,21]. This was in agreement with the results of SEM images as shown in Figure 4B,C. When force is applied, the molecular deformation movement is restrained, and this serves to reduce the fracture toughness [21,32].

On the other hand, no significant improvement in the impact strength was observed following the incorporation of either TiO_2_ or ZrO_2_ nanoparticles into the PMMA. More specifically, although the impact strength of the ZrO_2_ or TiO_2_ nanoparticle reinforced groups typically increased at 1.5 wt.% or 3 wt.% concentrations, the impact strength of the groups reinforced with 5 wt.% or 7 wt.% was reduced with insignificant differences. These results agreed with a previous study by Saleh et al., which found that the use of ZrO_2_ nanoparticles to reinforce high-impact PMMA has an insignificant effect on its impact strength [32]. In particular, they found that an increase in the concentration of ZrO_2_ gradually reduced the impact strength of the PMMA, with the exception of the 5 wt.% group. This outcome can be attributed to the fact that the incorporation of nanoparticles in the matrix may impede the development of links between the polymer chains, thereby reducing the rate of polymerisation [52], diminishing the quality of the chemical reaction that takes place between the PMMA and the particles at the interface, or leading to an inhomogeneous distribution of particles that leads to the development of particle clusters [32,50]. There is also a risk that incorporating nanoparticles into the PMMA can enhance the brittleness of the specimens, thereby reducing the overall impact strength [32]. The nanoparticle that had the greatest effect in terms of reducing the impact strength of PMMA was TiO_2_ at the higher concentrations of 5 wt.% and 7 wt.% where the strength was even lower than the control group. This could be attributed to the fact that TiO_2_ nanoparticles are particularly small [52]. Thus, they can more readily penetrate the space between the polymer molecules, are not easily dispersed in organic solvents, and typically agglomerate easily, undermining the mechanical properties of the material [52].

Previous studies have yielded conflicting outcomes in terms of the ideal concentration of fibres. Clarke et al. [35] concluded that fillers should be incorporated at a 2 wt.% fibre concentration. Gutteridge [15] tested 1 wt.% and 3 wt.% concentrations of ultra-high molecular weight polyethylene fibres and concluded that manipulation became more challenging when the ratio of fibres exceeded 4 wt.% in concentration. He recommended that for maximising the advantages of the fillers, they should be incorporated at a concentration of 1 wt.% [15]. Conversely, Karacaer et al. [17] tested 1 wt.%, 3 wt.%, and 5 wt.% concentrations of E-glass fibres within PMMA. They concluded that an increase in the concentration of fibres within injection-moulded specimens led to an increase in fracture toughness by approximately 35% and an increase in impact strength by approximately 34%. Furthermore, the highest values were observed when a fibre concentration of 5 wt.% was incorporated in the PMMA denture resin [17]. Likewise, Hamza et al. [7] found that incorporating different types of fibres (12-mm lengths) into PMMA enhanced its fracture toughness. However, Karacaer et al. [17] concluded that an increase in the fibre concentration led to an increase in the impact strength, but the effect of the E-glass fibres was not significant when tested on compression-moulded material. Finally, Vallittu et al. [53] found that the use of metal wires or E-glass fibres to reinforce acrylic resin did not significantly increase its impact strength.

The findings of the current study revealed that the use of E-glass fibre to reinforce PMMA denture significantly enhanced the fracture toughness of the denture base at all tested concentrations. Furthermore, the impact strength of the specimens in the PMMA group that was modified with E-glass fibre was higher than that of the specimens in the control group, with the exception of the E1.5 group. The largest increase in impact strength and fracture toughness was observed in the E7% group. This could be attributed to the fact that E-glass fibres exhibit a high degree of resilience that enables them to absorb stress or shock without resulting in premature deformation [28]. The outcomes of the current study were aligned with the findings of previous studies by Hamza et al. [7], Karacaer et al. [17] and Dikbas et al. [10], all of which found that modifying PMMA with E-glass fibres led to a significant increase in the impact strength and fracture toughness of the denture resin. These increases could be attributed to the stress transfer between the weak polymer matrix and high tensile strength of the fibres [7]. The stronger the level of adhesion between the fibre and the matrix as a result of silanisation, the stronger the resultant material is [7,28]. As can be observed in Figure 4D, this strong adhesion was evidenced in the SEM image. The increase in strength and fracture toughness might also result from the homogeneity of the compound, the effective level of fibre impregnation within the monomer, robust contact of fibres with the resin, an optimal amount of fibres within the resin, or the combination of organic resin with the inorganic fibres [7,28].

Disparities in the findings of the current study and previous studies may be due to differences in the methods of polymerisation, the approach used to prepare the sample, filler content, testing approach, size of samples, and material composition [14,21,43].

Incorporating E-glass fibres into PMMA denture bases can significantly enhance both their fracture toughness and impact strength properties. However, a low concentration of ZrO_2_ or TiO_2_ nanoparticles into the denture base can increase the fracture toughness only. Clinicians could use these findings to make a denture base with the reinforced PMMA for extending its longevity resulting in a base that is more durable and resistant to cracking materials.

Although a detailed life cycle analysis (LCA) has not been carried out as a part of this work, it is likely that the incorporation of nanoparticles or fibre in PMMA would require additional resources and processing, which could have an increased environmental impact. However, as denture bases fabricated with the composite materials would be more durable and have a longer life, the use of nanoparticles or fibre would balance the environmental burden to some extent by reducing the need for frequent visits to dental clinics, thereby reducing the emissions associated with transportation and the costs of consumables for dentists.

## 5. Conclusions

Although this study had some limitations, the findings confirm that the incorporation of E-glass fibre into heat-cured PMMA has a significant positive impact on both its impact strength and fracture toughness particularly at all concentrations higher than 1.5 wt.% used in this study. The inclusion of TiO_2_ nanoparticles at a concentration of 1.5 wt.% and the inclusion of ZrO_2_ at concentrations of 1.5 wt.% and 3 wt.% resulted in higher fracture toughness than the pure PMMA. However, reinforcing PMMA with nanoparticles did not lead to a significant improvement in the impact strength. Furthermore, higher nanoparticles concentrations showed a tendency to cause particle agglomeration forming larger particles. Therefore, it can be concluded that E-glass fibre represents a viable material for reinforcing PMMA as it can make a significant positive impact on the PMMA material properties.

## Figures and Tables

**Figure 1 materials-14-04127-f001:**
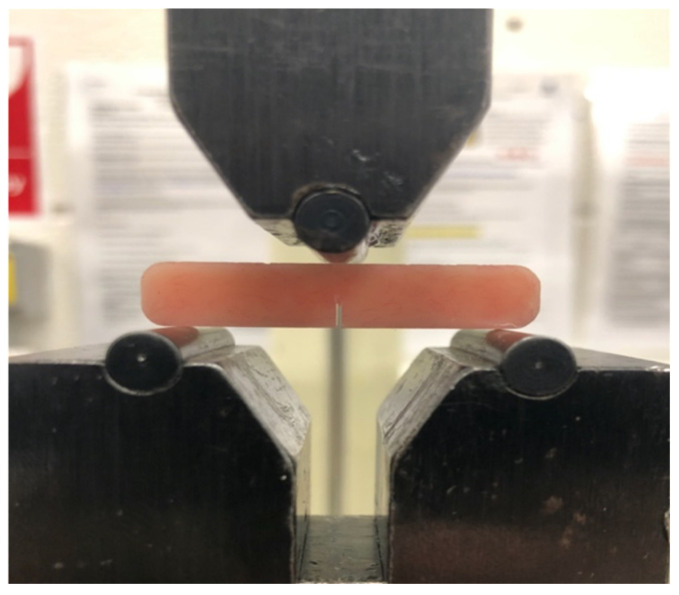
Single edge-notched specimen on the fracture toughness testing instrument.

**Figure 2 materials-14-04127-f002:**
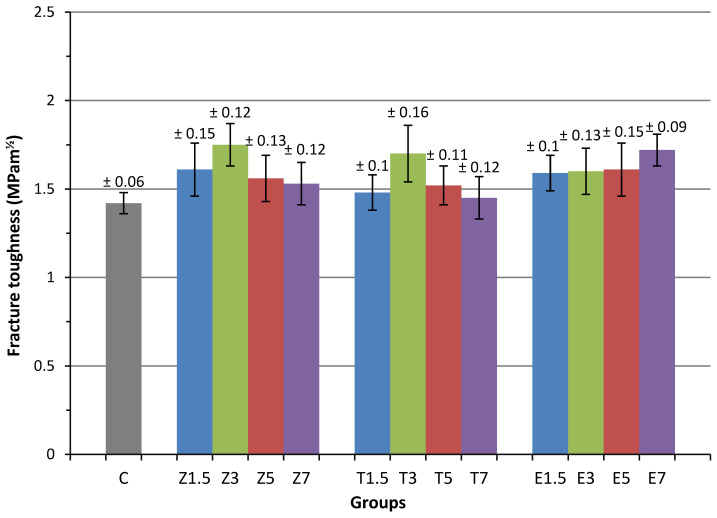
Mean fracture toughness of the reinforced PMMA specimen groups.

**Figure 3 materials-14-04127-f003:**
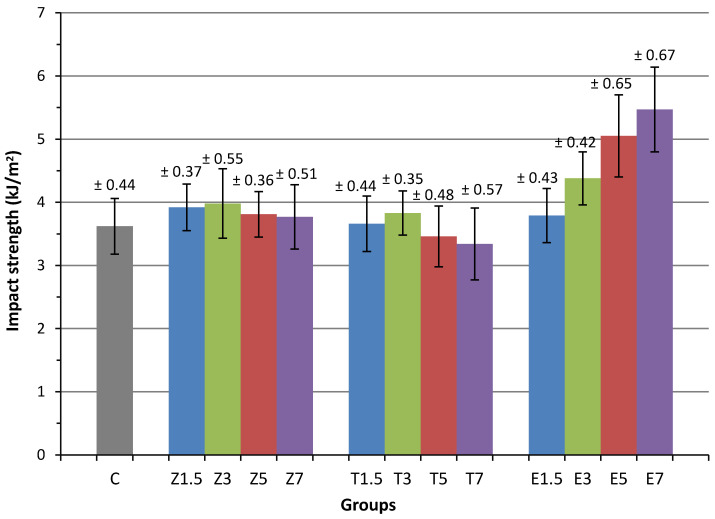
Mean impact strengths of the reinforced PMMA specimen groups.

**Figure 4 materials-14-04127-f004:**
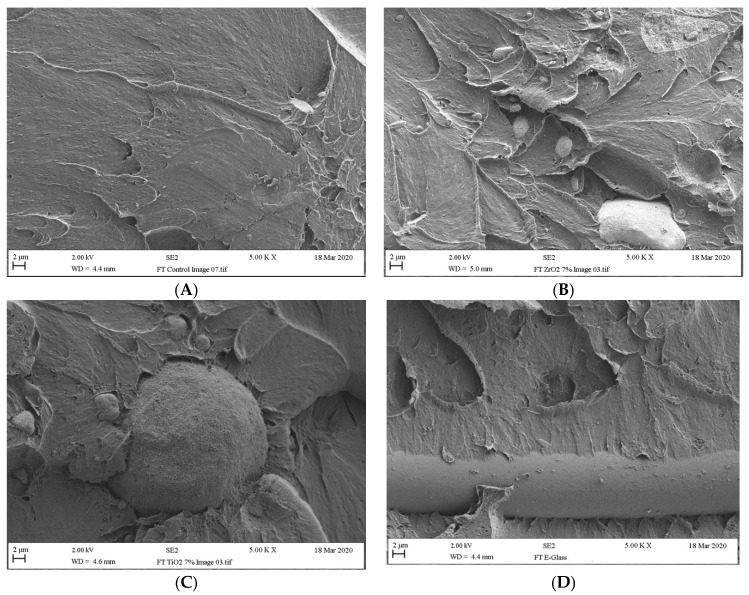
Fractured surfaces of (**A**) pure heat-cured PMMA; (**B**) ZrO_2_ reinforced composites; (**C**) TiO_2_ reinforced composites; (**D**) E-glass fibre reinforced composites.

**Table 1 materials-14-04127-t001:** Materials used for fabricating filler reinforced PMMA specimens.

Material	Composition and Specifications	Manufacturer
Lucitone-199^TM^	Heat-polymerised acrylic resin powder: PMMA; monomer: MMA	Dentsply International, York, PA, USA
Zirconium oxide	Zirconium (IV) oxide-yttria stabilised, nanopowder, <100 nm particle size	Sigma Aldrich, Gillingham, UK
Titanium oxide	Titanium (IV) oxide, anatase, nanopowder, <25 nm particle size	Sigma Aldrich, Gillingham, UK
Silanised E-glass fibre	3 mm in length, 15 μm in diameter	Hebei Yuniu Fiberglass, Xingtai, China
Ethanol	Ethanol, absolute (C_2_H_6_O, EtOH)	Fisher Scientific, Loughborough, UK
Silane coupling agent	3-(Trimethoxysilyl)propyl methacrylate, assay 98%	Sigma Aldrich, Gillingham, UK

**Table 2 materials-14-04127-t002:** Specimen grouping and coding.

Materials Groups	Filler Concentrations (wt.%)	Filler Concentration Subgroup Code	Material Description
Control	0.0	C	PMMA acrylic resin
ZrO_2_ nanoparticle	1.5	Z1	PMMA acrylic resin + 1.5 wt.% ZrO_2_
	3.0	Z3	PMMA acrylic resin + 3 wt.% ZrO_2_
	5.0	Z5	PMMA acrylic resin + 5 wt.% ZrO_2_
	7.0	Z7	PMMA acrylic resin + 7 wt.% ZrO_2_
TiO_2_ nanoparticle	1.5	T1	PMMA acrylic resin + 1.5 wt.% TiO_2_
	3.0	T3	PMMA acrylic resin + 3 wt.% TiO_2_
	5.0	T5	PMMA acrylic resin + 5 wt.% TiO_2_
	7.0	T7	PMMA acrylic resin + 7 wt.% TiO_2_
E-glass fibre	1.5	E1	PMMA acrylic resin + 1.5 wt.% E-glass
	3.0	E3	PMMA acrylic resin + 3 wt.% E-glass
	5.0	E5	PMMA acrylic resin + 5 wt.% E-glass
	7.0	E7	PMMA acrylic resin + 7 wt.% E-glass

**Table 3 materials-14-04127-t003:** Mean and standard deviation (SD) of fracture toughness and impact strength values for the tested groups.

Group		Fracture Toughness (MPa.m^1/2^) Mean ± SD	Impact Strength (kJ/m^2^) Mean ± SD
Control	C	1.42 (0.06) ^AD^	3.62 (0.44) ^ABC^
ZrO_2_	Z1.5	1.61(0.15) ^BC^	3.92 (0.37) ^A^
	Z3	1.75 (0.12) ^B^	3.98 (0.55) ^A^
	Z5	1.56 (0.13) ^AC^	3.81 (0.36) ^A^
	Z7	1.53 (0.12) ^AC^	3.77 (0.51) ^A^
TiO_2_	T1.5	1.48 (0.10) ^D^	3.66 (0.44) ^B^
	T3	1.70 (0.16) ^E^	3.83 (0.35) ^B^
	T5	1.52 (0.11) ^D^	3.46 (0.48) ^B^
	T7	1.45 (0.12) ^D^	3.34 (0.57) ^B^
E-glass fibre	E1.5	1.59 (0.10) ^E^	3.79 (0.43) ^CD^
	E3	1.60 (0.13) ^EF^	4.38 (0.42) ^D^
	E5	1.61 (0.15) ^EF^	5.05 (0.65) ^E^
	E7	1.72 (0.09) ^F^	5.47 (0.67) ^E^

Note: Similar superscript letters in the same column indicate no significant difference between each of the reinforced groups and the pure PMMA acrylic resin control group (*p* > 0.05).

## Data Availability

The data presented in this study are available within the article.

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
