# Peer review of "Assessing Fracture Toughness and Impact Strength of PMMA Reinforced with Nano-Particles and Fibre as Advanced Denture Base Materials"

_materials, 2021, doi:10.3390/ma14154127_

Round 1

Reviewer 1 Report

The article describes the effects of Eglass fibre, ZrO2 and TiO2 nanoparticles at different concentrations on the fracture toughness and impact strength of PMMA denture base. The major conclusion from the study is that the inclusion of E-glass fibres does significantly improve impact strength, while ZrO2 or TiO2 nanoparticles did not. It is an important study that could be used in other fields in addition to the dental industry.

Here are some major comments:

  1. When “PMMA” is mentioned, the authors have to write its full name.
  2. Did the impact strength was studied as a function of the size of E-glass fibres, ZrO2 or TiO2 nanoparticles? In the nanoscale its very important. If it wasn’t studied, at least the authers have to explain why they choose to use those nanoparticles' sizes and write what is mentioned in the literature about this parameter.
  3. The authors didn’t explain why they used those compositions of the nanoparticles. I think it is important. In addition, maybe there is a maximum addition in the composition of the nanoparticles; it is important to study this effect.

Author Response

Dear Professor, 

Please find a copy of respond attached. 

Best wishes, 
Abdulaziz

Reviewer 2 Report

1- Different techniques can be used for the toughening and making a balance between toughness and strength in Polymer based structures, including using of toughening agent, modification of the filler or reinforcement, and also sonication and annealing : please have a look at

  • https://www.sciencedirect.com/science/article/pii/S1359836821005059

2-in introduction you mentioned:

To date, a range of nanoparticles has been introduced to the polymer matrix, including ....,

Lots of important nanofillers have been missed: CNT, Graphene, GQD and also fibers including Carbon fiber.

3- In terms if sustainability and LCA, how do you rate your product?

4-Any data of thermal stability of the samples(TGA data)?

5- Any DSC, FTIR results?

3

Author Response

(The authors gave the same response as above.)

Round 2

Reviewer 1 Report

The authors responded to the comments×¥

Reviewer 2 Report

The new version of the manuscript may be ready for publish